# "The Subtle Craft of the Devil": Misogynistic Conspiracy Theories and the Secret Society of Pregnancy Cravings in E. T. A. Hoffmann's *Vampirism*

Michael Grant Kellermeyer

Independent Scholar, Fort Wayne, IN 46805, USA; mgkellermeyer@gmail.com

**Abstract:** This paper analyzes themes of male insecurities and distrust of the exclusive culture of female sexuality and reproduction in E. T. A. Hoffmann's *Vampirism*, one of the earliest psychologically sophisticated female vampires in Western literature. The doomed heroine, Aurelia, escapes a life of maternal abuse and sexual trauma by marrying the wealthy Count Hippolytus, but his attraction warps into suspicion when she becomes pregnant and loses her appetite for his food. Worried that losing her virginity has activated promiscuity inherited from her late mother, he begins following her and thinks he sees her conspiring with a coven of female ghouls who train her to satisfy her pregnancy cravings by feeding on a male corpse. Real or imagined, this vision confirms his suspicions and leads to their mutual destruction. In my analysis, I explore vampire literature's early history, its place within Gothic literature, the prominent role of female vampires, their relationship to gender anxieties exacerbated by the Romantic Era's subversive political movements, and the way in which Hoffmann's cynical story operates as a misogynistic conspiracy theory aimed at the secret female space of reproduction, symbolized by Aurelia's cannibalistic pregnancy cravings. As such, it contributes to the destructive folklore of social distrust.

**Keywords:** E. T. A. Hoffmann; Gothic literature; German Romanticism; pregnancy; cannibalism; vampirism; feminism





## 1. "Dangerous Sexuality": The Female Vampire in Early Gothic Literature

Prior to the advent of Bram Stoker's *Dracula*, the earliest literary vampires—an archetype that first rose to prominence in Europe during the late eighteenth century—were primarily female, hailing from a wide range of social castes; "a most diverse family" of characters that deviated widely from the now-dominant trope of the powerful male seducer (Holte 1999, p. 163). Female vampires remained in vogue into and throughout the nineteenth century, and their scandalous nocturnal occupations—like those of witches during the previous century—have widely been interpreted by scholars as an expression of the landed male establishment's anxieties surrounding female sexuality, including its mysteries, powers, exclusivity, and inaccessibility (Calzoni and Perletti 2015, p. 43). This fear of women's sexual influence—as from a place of great power—was of course paradoxical, as they were politically and socially disenfranchised, which has been regularly commented upon. Gender historian Gerda Lerner called the subject of female social influence "complex" and "full of paradoxes which elude precise definitions and defy synthesis" (Lerner 1979, p. 3), remarking that:

> Women at various times and places were a majority of the population, yet their status was that of an oppressed minority, deprived of the rights men enjoyed. Women have for centuries been excluded from positions of power, both political and economic, yet as members of families, as daughters and wives, they often were closer to actual power than many a man... The rationale for women's peculiar position is society has always been that their function as mothers is

> essential to the survival of the group and that the home is the essential nucleus of society as we know it". (Lerner 1979, p. 4)

Women's singular responsibility for gestating and giving birth to new human life gave them unique proximity to the deep mysteries of life and death, and the bloody, often life-threatening act of childbirth gave them a natural acquaintance with the ways of these liminal transitions that men only encountered in professions like medicine or soldiering.

Likewise, the vampire is a mysterious, metaphysical manipulator capable of crossing between the secret realms "between life and death and in a spiritual limbo betwixt heaven and earth" (Mulvey-Roberts 1998, p. 79). Feminist critics, including Pam Kesey and James Craig Holte, assert that "the image of the female vampire", an archetype "ancient in origin", was generated as an anxious response by "patriarchal attitudes" about the "destructive side—blood, death, and dangerous sexuality—of the great mother goddess of prehistory" (Holte 1999, p. 163). Indeed, the connection to the feminine is made all the stronger by its relationship to human blood, a fluid so closely associated with the female experiences of menstruation and childbirth. As Marie Mulvey-Roberts observes in her feminist critique of Dracula, the vampiric association with blood stems from widespread cultural taboos shared with women of child-bearing age: "[For] vampires, their victims and menstruating women, it is normal for blood to flow outside the body . . . transgressing the natural order, where blood is contained within the living body" (Mulvey-Roberts 1998, p. 79).

This cultural assumption that women cultivated a closer, almost supernatural relationship with the hazy borders between so many sacred spaces (life and death, sex and disease, birth and decay) arguably led to the proliferation of many gendered or eroticized supernatural tropes, including witches and vampires. Raul Calzoni asserts that "since their first scientific and literary appearances, vampires, like witches, have . . . been strictly connected with seduction and with the attempt of 'sexualizing death,'" observing that during "the Enlightenment the vampire became the substitute for the witch" becoming an embodiment for "diseases, plagues and epidemics . . . [in] localistic folklore" (Calzoni and Perletti 2015, p. 43). The *folkloric* vampire described in popular publications and rural legends was more akin to a ghoul or zombie—a "horrifying. . . ghastly" bloated cannibal whose stench, decomposition, and "gross corporeality" (Senf 2013, pp. 18–19) had none of the sleek sexuality of the alluring, Byronic bloodsuckers in high-brow literature, initiated by John Polidori's rakish Lord Ruthven, in 1816's The Vampyre (Senf 2013, p. 24). In his seminal survey of vampire lore, Vampires, Burial, and Death: Folklore and Reality, Paul Barber acknowledges that he:

> found it necessary to distinguish between the fictional and folkloric vampire. . . The fictional vampire tends to be tall, thin, and sallow, the folkloric vampire is plump and ruddy, or dark in color. . . The two would be unlikely to meet socially, for the fictional vampire tends to spring up from the nobility and live in a castle while the folkloric vampire is of peasant stock and resides (during the day at least) in the graveyard in which he was buried. (Barber 1988, p. 4)

These working-class thrillers began as highly localized ghost stories (Senf 2013, p. 19), but by the rise of Romanticism, the European bourgeoisie had sanitized, familiarized, and reappropriated the trope to dialogue with the far more tangible, domestic insecurities of the marriage bed. As M. M. Carlson notes, although "the origin of the literary vampire lies in folklore . . . literature has greatly reworked and remodeled the vampire into a recognizable literary type to suit its own needs and purposes" (Carlson 1977, p. 26). The taboos and gender-guarded secrecy of female arousal, seduction, and reproduction systems were analyzed in high-brow Gothic poems such as Johann Wolfgang von Goethe's *The Bride of Corinth* and Samuel Taylor Coleridge's *Christabel*, in which "[s]exuality begets violence, and male authority . . . is questioned by strong, intelligent, sexually active females", causing "monsters to be created out of perceived threats to patriarchal order" (Holte 1999, p. 164).

One example of early vampire literature that deserves a closer study into its particularly cynical interpretation of female sexuality and reproduction, comes from Ernest Theodor Amadeus Hoffmann, one of Germany's most significant authors, known for

writing "some of the most influential literary fairy tales in the Western cultural tradition" and for his indelible influence on modern horror and fantasy (Owen and Crawford 2020, pp. 13–14). The fourth volume of his 1821 novel, The Serapion Brethren, included an episode that has been published as an excerpt under various names, including *Tale of a Vampire*, *Cyprian's Narrative*, *The Hyena*, *Vampyrismus*, and (as we shall be referring to it) *Vampirism* (Kamla 1985, p. 235). In it, Hoffmann's suspicious treatment of pregnancy, and pregnancy cravings in particular, is an especially telling example of how nineteenth century vampire literature attempted to defang the tabooed "secrets" of female sexuality. Today, Hoffmann is best remembered for his surreal fantasies that were dark, Kafkaesque fairy tales enhanced by contemporary settings, proto-Freudian eroticism, and existential angst (Owen and Crawford 2020, p. 14). Most influential among these are *The Sandman*, *Nutcracker and the King of Mice*, *Councilor Krespel*, and *The Golden Pot*. These "provocative tales" were populated by psychologically realistic gnomes, elementals, witches, vampires, ghosts, doppelgangers, and animated dolls who often initiated the downfalls of the protagonists—idealistic loners whose violent breaks with reality blurred "the boundaries between reality and fantasy, evoking uncanny conflations . . . cajoling readers to reconsider their perceptions of reality and social convention" (Owen and Crawford 2020, p. 13).

Although *Vampirism* is not one of Hoffmann's major works, it has been analyzed by several excellent scholars. Thomas A. Kamla described it as "a manifestation of all too human aberrations which . . . reside 'within our own breast'" in his psychoanalytical interpretation (Kamla 1985, p. 235), Heide Crawford discussed its literary ethos as "a model proving that a horror story can be aesthetically pleasing" (Crawford 2009, p. 24), and Nicole A. Sütterlin argued that it simultaneously "explores the transgressive powers of literature to affect reality" and "contains a critique of . . . psychiatry" in her analysis of its subversive poetics (Sütterlin and Lambrow 2018, p. 116). Others, such as Horst S. Daemmrich, have lambasted its "nauseating[ly]" puritanical morality and lack of "artistic subtlety" (Daemmrich 1973, p. 87), while several critics, including James M. McGlathery, have explored its psychosexual implications as a Gothic tale in which the "connection between erotic passion and insanity" is "manifested in a young woman's hysteria" (McGlathery 1997, p. 106). Little if nothing, however, has been written about this story from a focused, feminist perspective, and none of the available literature has thoroughly addressed the way in which *Vampirism* demonizes pregnant women, or how its male protagonist arguably soothes his misogynistic insecurities by crafting a series of gendered conspiracy theories concerning the exclusively female culture of gestation. While some of the previous commentators—McGlathery and Daemmrich in particular—have read *Vampirism* with some feminist considerations, none of the present research has offered a committed interpretation of its gender dynamics. As such, this article aims to provide a much-needed critical contribution to its sadly relevant narrative of conspiratorial misogyny.

It is also noteworthy that while feminist critiques of vampire literature are exceedingly commonplace, the intersection of feminism and conspiratorial rhetoric *within* the vampire myth is not nearly as developed. Indeed, the literature surrounding conspiracy ideation continues to center around Marxist and populist jealousies of the established power holders, focusing on "high levels of suspicion towards centralized authority and their political elites" (Oliver and Wood 2014, p. 952), and the tendency of conspiratorial ideation to be most "prevalent among members of low-status groups attempting to explain their status" (Douglas et al. 2019, p. 9). This downplays the increasing influence of *downward*-punching folklores of social distrust, the aims of which are to "justify radical, exclusionary politics" and further disenfranchisement of vulnerable communities by casting them as dangerous usurpers, "casting even moderates as part of a conspiracy" and minorities "as a potent threat to civic life" (Douglas et al. 2019, p. 14). Conspiracy theories that perpetuate the devaluation or dehumanization of socially aspirant communities continue to be "abundant in social and political discourse", as they are passed through popular culture and perpetuated through literary tropes (Douglas et al. 2019, p. 3). The paranoid visions in *Vampirism* belong to this second group and can be placed within the broader cultural

context of early Gothic literature, the social turmoil of the French Revolutionary Wars, and the regressive anxieties of the European bourgeoisie amidst an era of surging progress. In this paper, I will explore how Hoffmann's vampire tale employs the female vampire archetype to construct a literary conspiracy theory founded on misogynistic distrust of the perceived secret, exclusively female culture surrounding female reproduction. By analyzing this paranoid dialectic, we will be able to better understand how conspiratorial ideation and the folklores of social distrust are often invented and disseminated by cultural majorities in a jealous bid to alienate and "other" vulnerable communities—women among them—perceived as threats to their power monopolies.

## 2. "All Kinds of Envious Forces": Gothic Conspiracy Theories of Dissident "Others"

At the time when Hoffman began writing, Gothic fiction was experiencing a surge in cultural currency in Germany, which fueled a "pan-European phenomenon . . . popular on an international scale", inspired by Immanuel Kant's philosophies on sublime aesthetics to create a unique literary tradition (Crawford 2009, p. 19). As one of the movement's preeminent voices, Hoffmann created a distinctive Gothic aesthetic "in the way it demands that truly gruesome horror stories be . . . based in reality or realistic situations" (Crawford 2009, p. 20). The Gothic sensibilities with which we now associate horror fiction were first formulated in European literature during the jolting shift from the Enlightenment (an era of optimistic collectivism and social development) into a period marked by staggering social upheaval and political division, most notably instigated by the chaotic French Revolution, for which the Gothic "did in fact serve as a metaphor" (Paulson 1981, p. 534). Carlson directly connects the vampire to the philosophical moment of the late eighteenth century—the transition from the Enlightenment's elitist optimism to Romanticism's populist contrarianism:

> The modern literary vampire first made its appearance in a period of reaction against the domination of rationalism. Vampire literature per se is a post-Enlightenment phenomenon, a result of Romanticism and of the interest in folklore encouraged by Herder's romantic nationalism. Through German Romanticism, the vampire made his way from folklore into the realm of literature. (Carlson 1977, p. 26)

One of the Gothic movement's recurring literary motifs, closely tied to the eroding social trust of a continent submerged in unending imperial wars, was the prevalence of conspiracies among classes of people whom readers would deem suspicious, namely the polar margins of European society, with the social elite on one side and disenfranchised castes (e.g., women, foreigners, Freemasons, Jacobites, and religious minorities) on the other (Paulson 1981, pp. 540–41). Postcolonial scholar Tabish Khair begins his treatise on Gothic literature with the assertion that "[t]he Gothic and the postcolonial are obviously linked by a common preoccupation with the Other and aspects of Otherness" (Khair 2009, p. 3), clarifying that "[w]hen I suggest that Gothic fiction is a 'writing of Otherness,' I allude most simplistically that it revolves around various versions of the Other, as the Devil or as ghosts, as women, vampires, Jews, lunatics, murderers, non-European presences, etc." (Khair 2009, p. 5). Marxist critic Frederic Jameson concurs that:

> Gothics are indeed ultimately a class fantasy (or nightmare) in which the dialectic of privilege and shelter is exercised: your privileges seal you off from other people, but by the same token they constitute a protective wall through which you cannot see, and behind which therefore all kinds of envious forces may be imagined in the process of assembling, plotting, preparing to give assault. (Jameson 1991, p. 289)

Regardless of their political affiliation, conspiracy theories can be described as "narratives about hidden, malevolent groups secretly perpetuating political plots and social calamities to further their own nefarious goals" (Oliver and Wood 2014, p. 952), which "may provide a more accessible and convincing account of political events" while "draw[ing] heavily upon

the idea of unseen, intentional forces shaping contemporary events" (Oliver and Wood 2014, p. 964). As Europe descended into international chaos, the local literature responded with conspiratorial narratives, among which sinister tales of scheming cabals may have been oddly comforting to mainstream readers desperate to knit order out of confusion.

The English novels that ushered in Gothic literature tended to create supervillains by merging marginal castes. The antagonists in works such as The Monk, Melmoth the Wanderer, and Vathek were almost always foreign religious minorities (Catholics, Jews, pagans, or Muslims) harboring morally alarming secrets related to their sexual and spiritual degeneracy (Khair 2009, p. 6). However, these early books still largely operated out of a rationalistic, Enlightenment sensibility, explaining *away* supernatural events as either carefully designed frauds used to cover up elaborate conspiracies, or as flights of fancies primed by the overactive imaginations of gullible non-Protestant characters, allowing "middle class readers, safely tucked into their stable and unthreatened the social positions, [to] feel secure enough to cultivate imaginary fears and fantasies . . . while apparently immune from real danger" (Stevens 2000, p. 10). Hoffmann, on the other hand, had far fewer qualms about depicting supernatural phenomena. Indeed, as Crawford notes, this was a common feature of German Gothic literature at the time, which fostered stories with uniquely surreal, fairy tale qualities:

> Toward the end of the eighteenth century, the German Gothic horror literature trend began to develop in a different direction that its British counterpart, due, in part, to the influence of Kant's ideas concerning aesthetics and the sublime on German authors of horror. These authors were less concerned than their British counterparts with providing rational explanations for the perceived supernatural; they were interested in creating a lasting and continuous impression of sublime horror fraught with mystery. (Stevens 2000, pp. 19–20)

Hoffmann's *Vampirism certainly* refuses to provide a comforting, rational explanation, nor, however, does it definitively authenticate its supernatural narrative. Instead, it requires the reader to suffer in a liminal no-man's land, questioning the possible reasons of a hallucination or possible implications of a veridic supernatural event (Crawford 2009, p. 24). The story would prove to be simultaneously remarkable, for its psychological nuance, and conventional, for its employment of the Gothic dialectics of Otherness—particularly the trope of the vampire. By employing this increasingly popular stock character, Hoffmann's story positions itself as being concerned with social upheaval and subversive conspiracies. Even modern vampires who are treated as misunderstood *Übermenschen* on the margins (cf. the pointy-toothed anti-heroes Lestat, de Pointe du Lac, Spike, Angel, Edward Cullen, and Blade) operate as countercultural actors, resisting conventionality and swimming against the societal mainstream. These so-called "'post-vampires'"—straight-edge vegans, humanitarians, and abstainers—defiantly "re-work traditional conceptions of the supernatural figure" in a time of unchecked decadence and appetite (Park and Wilson 2011, p. 3). By their very nature, vampires (including Hoffmann's) are commonly suspected to be motivated by a desire to secretly leech power from their culture's gatekeepers—literally when they stealthily feed off of the beautiful victims whom they target for their vitality, and metaphorically when they corrupt youths, seduce married women, scandalize the bourgeoisie, blaspheme the Church, infect the living, and generally spread their contagion throughout whichever country they have opted to convert to their nocturnal way of living. Barber points out that, despite the slothful appearance of their "uncomplicated life" spent "quietly in [their] castle", only rising from their heavy, prolonged slumber to feed at night:

> [t]he vampire is not without energy and purpose. . . Often enough, [they are] shown to be engaged in an effort to do nothing less than take over the world, with the aid of an army of subordinate vampires. If one excepts his craving for blood, this power-lust is his sole passion and is seldom explained or analyzed. To be a vampire, it seems, is to be power-mad, in the grip of a compulsion. . . (Barber 1988, p. 183)

This correlation between vampires and secret political cabals is supported by the psychological literature on conspiracy theories, which demonstrate that "'conspiracist ideation' . . . is actually derived from [a] psychological predispositions. . . to attribute the source of unexplained or extraordinary events to unseen, intentional forces. . . often found in supernatural, paranormal, or religious beliefs" (Oliver and Wood 2014, p. 954). As turn-of-the-century Europe experienced a seemingly spontaneous surge in revolutions, coups, reforms, and social turmoil, it is perhaps unsurprising that the mainstream bourgeoisie, distressed and confused by the surge in threats to their powerful allies, would begin to generate narratives about power-hungry conspirators emerging from the ranks of the marginalized and disaffected classes, including unorthodox intellectuals, minorities, and women. As we shall see, the female bloodsucker in Hoffmann's tale is at the very least placed in a posture of power hunger, although his subtle characterization leaves much room—arguably in spite of his own prejudices—for a far more nuanced analysis of just who it is in this story that is coveting power from whom.

Despite its role as an influential pioneer in the vampire genre, or the psychological complexity of its star-crossed protagonists, the story has failed to attract the sort of critical attention one might assume. Kamla agrees that the story has "suffer[ed] as a work of literature, being regarded by most critics . . . as a popular [*viz.*, pedestrian] contribution to Gothic horror", pointing to the seemingly tactless "dehumanized instinctual impact of the conclusion", which has led to its general dismissal and a lack of "serious, in-depth investigations" of its merits, which he notes "are virtually nonexistent" (Kamla 1985, p. 237). Nonetheless, he makes a convincing argument that this assessment has been shortsighted—that it is "deserving of such concerns . . . if for no other reason that that a latent content informs the text which makes the closing scenes—the visual presentation of this content on the manifest level—psychologically (per Cyprian) explainable." Indeed, like Hoffmann's two most famous works, *The Sandman* and *Nutcracker and the King of Mice*, it is easy to mistake *Vampirism* for a schlocky children's shocker, overlooking its rich literary depth. While *Sandman* and *Nutcracker* have had their influential champions since the days of Sigmund Freud and Alexandre Dumas (respectively), *Vampirism* now demands a reassessment—especially during an age that must be increasingly concerned with understanding the cultural narratives of misogyny that motivate the distrust and demonization of women as insidious keepers of secrets.

### 3. "Accursed Misbirth of Hell": A Summary of E. T. A. Hoffmann's *Vampirism*

In the tale's framing narrative, we listen in on a meeting of the Serapion Brothers—a literary society of Prussian writers and intellectuals. One member, Sylvester, brings up Polidori's recent, sensational novel *The Vampyre*, which had electrified post-war Europe with its grisly details and unvarnished cynicism. At Sylvester's prompting, the group begins discussing what odds and ends they know about vampire mythology—a topic they consider repulsive and deeply disturbing. The friends' contributions include two quotations from historically reported cases of vampirism (a 1725 treatise from the Rev. Michael Ranft and a 1732 letter from Lt. Sigismund von Kottwitz), before one of them, Cyprian, offers his own true story (or so he has been told) about a vampire. He had "either heard or read a very long time ago" but cannot recall which, although he has a vague memory of being told the real name of the family and the estate where it took place (Hoffmann 1892, p. 455). In any case, he warrants that it is truly "ghastly."

It follows the rise and downfall of the beautiful Aurelia, a debutante "almost starving, in the depths of poverty" with her mother, a widowed baroness (Hoffmann 1892, p. 457). One day, they learn that their distant relation, Count Hyppolitus, has inherited his late father's massive estate, and decide to congratulate him in hopes of reversing their situation. Hyppolitus remembers that his father despised the baroness, though he cannot remember why, and although he is enchanted by the graceful Aurelia, he cannot deny that something about her elderly mother—with her bony frame, corpse-like pallor, and cloudy, blind eyes—is decidedly repulsive. He is most stunned when, in the act of shaking her hand,

she crushes his hand ("as if by a vice") with her ice-cold fingers. Conscious of his shock, Aurelia immediately apologizes, explaining that her mother is occasionally overtaken by "tetanic spasms." Her soothing intervention prevents Hyppolitus from turning them out, and he invites them to stay with him as honored guests. Although he never loses his disgust for the baroness—being particularly disturbed by the servants' rumors that she secretly walks to the graveyard each night—the Count becomes obsessed with Aurelia. He loses control over his intuition and is overwhelmed by "the full force of passion, so that it was impossible for him to hide how he felt" for Aurelia (Hoffmann 1892, p. 458). Before long, they become engaged.

Their wedding, however, is almost immediately marred by scandal when the baroness is found dead near the cemetery that morning, throwing Aurelia into a hysterical panic attack during which she admits to her husband that his father—who had narrowly avoided becoming engaged to her mother—was right about her mother's character as she was an *astonishingly* abusive and promiscuous woman. She shares multiple childhood traumas with him, beginning with a childhood memory of being shown her father's corpse laid out on a table, running up to it and being filled with a sudden urge to kiss the supine man, only to be horrified by finding that "his lips, always warm before, were cold as ice" (Hoffmann 1892, p. 461). Soon after, she is collected by a strange woman in a carriage whom the servants call her mother—the baroness. The rest of her childhood is a blur, until the age of sixteen when her mother began to be visited by a savage suitor whom she only refers to as "the Stranger", a purported baron who begins paying their bills and buying them luxurious presents. But his visits terrorize Aurelia, who detests his leering gaze and lecherous behavior, and she is proven right when she barely escapes being raped by him, only to be scolded by her mother for not repaying his largesse with sexual favors.

One night Aurelia attempts to escape the house, but as she nears the door, it flings open to reveal her soiled, half-naked mother and the Stranger, who, we are led to understand, has been pimping her out around town in exchange for financial assistance. Her mother staggers inside, and the furious Stranger begins dragging her around by the hair, pummeling her almost to death. Overwhelmed, Aurelia calls for help and a crowd assembles. In the process of arresting him, it is discovered that the Stranger is a con man. In fact, he is the executioner's son, a violent criminal who is wanted for theft and murder. Far from being grateful for her bravery, however, the baroness brutally punishes Aurelia for having scared away their benefactor and begins locking her up in her room all day until they befriend Hippolytus. Even this positive turn of events is spoiled by the baroness' hateful abuse; upon learning of the engagement—perhaps jealous that Aurelia has succeeded where she failed with Hippolytus' father—she puts a curse on her daughter:

> You are my misfortune, horrible creature that you are! But in the midst of your imagined happiness vengeance will overtake you, if I should be carried away by a sudden death. In those tetanic spasms, which your birth cost me, the subtle craft of the devil.—(Hoffmann 1892, p. 466)

But Aurelia is too traumatized to finish the curse, only saying that she genuinely fears that her mother might climb out of her grave to drag her away from her happy marriage and into hell. Exhausted and embarrassed, she tries to backtrack her anxieties and attributes her mother's cruelty to "the delirium of her insanity."

However cathartic her story may have been, Hyppolitus notes that from that point on, she seems to grow even more secretive and anxious, as if she were responsible for hiding some dreadful thing. She begins taking lengthy, lonely walks, and growing increasingly agitated:

> In a very short time Aurelia began to alter very perceptibly. Whilst the deathly paleness of her face, and the fatigued appearance of her eyes, seemed to point to some bodily ailment, her mental state—confused, variable, restless, as if she were constantly frightened at something—led to the conclusion that there was some fresh mystery perturbing her system. She shunned her husband. She shut

herself up in her rooms, sought the most solitary walks in the park. (Hoffmann 1892, p. 466)

Distraught, Hyppolitus summons a renowned doctor who informs him that, while he cannot definitively diagnose her malaise, he deduces that she is pregnant: "the happy result of a fortunate marriage" (Hoffmann 1892, p. 466). The doctor then goes on to aimlessly wax on about the strangeness of pregnancy cravings and the many "wonderful longings which women in that condition become possessed by, and which they cannot resist without the most injurious effects supervening upon their own health" (Hoffmann 1892, p. 467). Sometimes, he notes darkly, while Aurelia listens with unusual attentiveness, "these strange, abnormal longings" can lead to tragedies, even "terrible crimes." For example, he tells them about the disturbing case of a pregnant woman who began to foster a psychotic desire to eat her husband's flesh. One night, when he came home drunk, she killed and butchered him with a carving knife. The story has a violent effect on Aurelia, who collapses in a series of "hysterical attacks" before being revived by the apologetic physician.

The couple try to be happy with their good news, but new developments only increase Hippolytus' concerns. Aurelia grows more ghostly, with "a darksome fire in her eyes", an increase in her "deathlike pallor", and—strangest of all—she has ceased eating entirely: "she never took the smallest morsel of anything to eat, evincing the utmost repugnance at the sight of all food, particularly meat" (Hoffmann 1892, p. 467). Soon one of Hyppolitus' servants comes forward with disturbing information that only complicates the expanding mystery: every night since the wedding, Aurelia has been slipping out of the chateau, only returning just before daybreak. Hyppolitus, who has been sleeping unusually heavily since then, wonders if she is drugging him in order to see a lover like her lecherous mother:

> The Count's blood ran cold... The darkest suspicions and forebodings came into his mind. He thought of the diabolical mother, and that, perhaps, *her* instincts had begun to awake in her daughter. He thought of some possibility of conjugal infidelity. He remembered the terrible hangman's son. (Hoffmann 1892, p. 468)

His fears are at least partially accurate. That evening, he discreetly avoids drinking the tea she makes him each night and is disturbed to find himself alert when he would usually be overwhelmed with sleepiness. At midnight, while spying on his wife, he sees her steal out of the chateau, making a beeline through the countryside to a local graveyard. Following her through the gloomy moonlight, he is mortified to find a violated grave surrounded by a ghoulish coven of cannibals:

> There, in the bright moonlight, he saw a circle of frightful, spectral-looking creatures. Old women, half naked, were cowering down upon the ground, and in the midst of them lay the corpse of a man, which they were tearing at with wolfish appetite. Aurelia was amongst them. (Hoffmann 1892, p. 469)

Stunned, he staggers around the countryside until dawn, but when he returns to the chateau, Aurelia is peacefully asleep in bed, and he is relieved by the hope that it was all just a gruesome dream. Notwithstanding, when they sit down to breakfast and she refuses the food from his table as usual, he is overwhelmed with violent emotion, calling her an "accursed misbirth of hell" and revealing that he knows that she has been getting her "sustenance out of the burying-ground" (Hoffmann 1892, p. 470). Instead of being shocked, hurt, or embarrassed, sweet, docile Aurelia lunges at him:

> As soon as those words had passed his lips, the Countess flew at him, uttering a sound between a snarl and a howl, and bit him on the breast with the fury of a hyena. He dashed her from him on to the ground, raving fiercely as she was, and she [expired] in the most terrible convulsions.

Shaken to the depths of his soul by this sight, Hyppolitus is driven hopelessly insane.

At the closing of the tale, Cyprian's friends express shock and disgust, calling Ranft and von Kottwitz's stories nursery room thrillers by comparison. However, one of them—Theodor—quietly thanks Cyprian for censoring himself. He had read a *fully* detailed account of Aurelia and her mother years ago and particularly remembers the baroness'



"atrocities . . . in all their minutae" ([Hoffmann 1892](), p. 470). Traumatized by the memory of those details, he quietly admits that it "took a long while [for him] to get over."

### 4. Madonnas to Whores: Misogynistic Distrust in *Vampirism* and Hoffmann's Life

Despite this story's cool critical reception, its influence on the modern vampire story has been significant, particularly in its native milieu, where it established "a horror aesthetic that was uniquely German" ([Crawford 2009](), p. 20). The very earliest literary vampires appeared in Gothic-themed Romantic poetry (e.g., John Keats' *La Belle Dame Sans Merci*, Gottfried August Bürger's *Lenore*, Coleridge's *Christabel*, and Goethe's *The Bride of Corinth*) which consistently "look[ed] back to the Middle Ages or classical Greece" for their settings and inspiration ([Senf 2013](), p. 24). However, Hoffmann's socially realistic, psychologically nuanced treatment of the vampire continued Polidori's departure from convention, arguably helping to cement it as a viable trope for modern literary fiction. His cutting-edge use of magical realism—"where the realms of fantasy are continuously encroaching and populating the realms of the real" ([Owen and Crawford 2020](), p. 14)—popularized the now-universal concept of vampires as complex, modern characters rather than as mindless monsters restricted to mythology (*à la* ogres, dragons, or trolls). By placing the story within a contemporary setting (quite unlike his predecessors' distant medieval worlds) and by emphasizing the unqualified uncanny rather than overexplained supernatural, his multifaceted and relatable Aurelia debatably shares more with the likes of Emily Brontë's doomed outsider, Catherine Earnshaw, than a melodramatic villain in one of the Grimms' fairy tales.

Hoffmann's own unique take on the vampire myth—wherein an aristocratic female passes as a conventional ingénue, is welcomed into a trusting home, and proceeds to infect her hosts and the surrounding community with her unorthodox degeneration from female sexual mores—was truly original at the time and bears unpacking. Its influence on J. Sheridan Le Fanu's vampiric magnum opus, *Carmilla*, and Bram Stoker's *Dracula*—two works that "vampirically reenact the literary tradition started in Germany"—is obvious ([Calzoni and Perletti 2015](), p. 60). Although the story is not as well-known in the Anglosphere as those of Polidori, Le Fanu, or Stoker, for post-Enlightenment Germans, it was regarded as something of a watershed in German literature as a synthesis of aristocratic tastes and populist morals ([Crawford 2009](), p. 24). Crawford contends that "Hoffmann was consciously taking on the challenge of developing a model for a horror aesthetic . . . by including a rather gruesome vampire story among the tales the [Serapion] brethren tell each other" ([Crawford 2009](), p. 21) and that he intentionally crafted it in a "tasteful manner" meant to prove that "a horror story can be aesthetically pleasing" ([Crawford 2009](), p. 24). What was once the province of working-class folklore had been reformed, repackaged, and successfully commercialized for the gentry, but its core message—one of distrust for women and of male anxieties surrounding the uniquely female mysteries of procreation—would experience no such cosmopolitan rehabilitation. Instead, it represents Hoffmann's own boorish view of women, one framed by his passionate subscription to the Madonna/Whore dichotomy, and a staggering lack of moral complexity for a man of so much evident imagination, for, as Daemmrich critiques:

> The motif [of cannibalism and vampirism] which recurs in *The Serapion Brethren* is not only nauseating but also lacks artistic subtlety. It certainly reveals Hoffmann's thinking. Pure evil gains immortality through the sacrifice of innocence. Those who encounter it innocently suffer the most horrible destruction". ([Daemmrich 1973](), p. 87)

Like the very trope of the vampire itself, Hoffmann's story expresses a deeply conservative anxiety narrative about the infiltration and covert transgression of moral orthodoxy by an infectious strain of Otherness. Discussing the symbolic significance of Hoffmann's telling choice in monster, Sütterlin argues that:

> At the heart of [Hoffmann's] vampire poetics lies the idea of transgression. As a paradoxical corporeal specter and a figure who is both non-living and non-dead,

> the vampire not only transgresses the fundamental dichotomy of life and death, but transgression and paradox form this figure's very essence... Hoffmann's "Vampyrismus" tale is not only rife with paradox ... but it illustrates how the metaphorical can transgress into the literal and how fiction can become reality. (Sütterlin and Lambrow 2018, pp. 116–17)

The vampire was the ideal mythic vessel for Hoffmann's gender parable, a cautionary tale in which Aurelia is cast as a subversive insurgent intent on undermining the social order represented by Hippolytus and his noble family. Literary vampires are frequently deployed by their authors as disruptive dissidents intent on undermining the hierarchies of the societies they infiltrate (Holte 1999, pp. 163–64). This is why, as many commentators have noted, vampires are almost universally drawn from marginal social castes (promiscuous women at the turn of the eighteenth century, wealthy foreigners at the turn of the nineteenth, and teen delinquents at the turn of the twentieth), for as Khair asserts: "the Vampire myth depends on various discourses and fears that were most commonly employed to deal with colonial Otherness", specifically designed to generate pedagogical "discourses" against rising threats to the existing social order, "for instance, the 'depravity of women' or the post-1789 fear of the violent revolutionary masses who were often shown as cannibalizing the better classes" (Khair 2009, pp. 57–58). As Barber wryly observes, "[i]n general, lists of potential revenants tend to contain people who are distinguished primarily by being different from the people who make the lists (Barber 1988, p. 30).

Upon a first reading, the insidious "Other" in question initially appears to be Hyppolitus' unstable mother-in-law, with her rumored criminal history, "repulsive face", muscular hands ("cold and stiff as death"), and "generally spectral appearance" (Hoffmann 1892, pp. 457–58). We are quite deep into the story before we suspect that the docile beauty Aurelia is the titular monster. Her shift from a compliant dependent to a rebellious temptress serves as the crux of the plot's horror, and her transmutation from the "noble, simple, and child-like" (Hoffmann 1892, p. 458) girl, who so easily disarms Hippolytus' intuitive distrust of his future mother-in-law, to a hysterical cannibal can be read as a cautionary tale warning against the conspiratorial machinations of women. As McGlathery opines: "[Hyppolitus'] ensuing fantastic experiences as Aurelia's husband may be seen to confirm his unconscious worst fears about surrender to desire for women" (McGlathery 1997, p. 107). The very nature of Aurelia's starved vampirism—one motivated by the burning hunger of a "wolfish appetite" (Hoffmann 1892, p. 469)—speaks volumes about the threat she represents to Hyppolitus' patriarchal authority. Rather than slowly slurping up the blood of sleeping victims, she is seen inhaling mouthfuls of human flesh from "the corpse of a man [emphasis added]", whose bodily sovereignty has been given over to this young woman as she is coached in cannibalism by a group of female elders. Kamla observes that:

> From the outset, the title 'Vampirism,' of any of its variants, needs to be qualified, as it conveys an inadequate message for the kind of pathological behavior portrayed in the story ... the oral aggression exhibited by ... [Aurelia] extends beyond the infantile urge to suck ... blood, and instead combines the urge to bite with that of tearing and devouring. It is really ... ghoulish cannibalism. (Kamla 1985, p. 236)

The threat she and her mentors pose to males is more holistic than the traditional model of a vampire who gradually weakens and influences her victim. Instead, we find a conspiracy to completely engulf and dominate the bodies of men.

This misogynistic distrust appears to have a biographical source in Hoffmann's *own* sexual frustrations. As Daemmrich observes, Hoffmann's frequent depictions of "struggle between artist and temptress" could be reflective of "[his] personal tragedy" with a number of humiliating romantic rejections—most famously that with his fourteen-year-old music student, Julia Mark (Daemmrich 1973, p. 34). This infamous obsession caused the thirty-four-year-old married Hoffmann to lose trust in his own senses and body, leading to very real "fears that he himself might be losing his mind or be driven to suicide" (McGlathery 1997, p. 11). Eventually, Mark's watchful mother was so disgusted by his obvious attraction

to her teenage daughter—a lust that culminated in an "inappropriate" gift of "erotically tinged verses he wrote to the daughter for her 15th birthday"—that she angrily dismissed him, but not before his fanatical lust had metastasized into "an emotional crisis . . . [taking] on the character of a jealous passion" (McGlathery 1997, p. 12). Publicly rebuffed by the mother and quietly ignored by the daughter, he would go on to interweave his fiction with pejorative female tropes, including the Whore and Virgin, the teasing temptress, the deceitful gold digger, the ugly hag disguised as a beautiful seductress, the withholding prostitute managed by a taunting pimp, and women from different generations (daughters and mothers, maidens and matrons) conspiring together to humiliate adoring suitors. As Daemmrich notes:

> The symbolism of the virgin and the vampire constitutes one of the most elaborate forms of doubling in Hoffmann's works. It occurs in many stories. Sometimes the women are easily identified because the hero . . . is confronted by a clear choice between a pure, good woman and a vampire . . . In [other tales] the characteristics of the women remain veiled until they engage in an almost dramatic struggle . . . The symbols, however, belong to an established tradition. The choice of the untouched woman has religious overtones in recalling the cult of the Virgin Mary, while the vampires are reminiscent of Lilith and the Whore of Babylon. In Hoffmann's tales the symbolism of the virgin and the vampire reflects the struggle between the ideal and reality, between the angelic and satanic vision in man. (Daemmrich 1973, pp. 34–35)

Hoffmann's life and fiction were both spent pondering the inevitable degeneration of young unspoiled girls into crafty heartbreakers at best, and demonic prostitutes at worst (cf. *The Empty House*, *The Artus Exchange*, *The Jesuit Church in G.*, *The Story of the Lost Reflection*, *The Sandman*, etc.). On the other hand, his typical male protagonist's "innocent, almost pious love" causes him to be "overpowered by his passion" (Daemmrich 1973, p. 34). Eventually, this humiliated lover "awakens one day and realizes that the beautiful girl is in reality a strange, seductive temptress who possesses his body and soul." A charitable biographer might allow that his adverse childhood experiences with an emotionally abusive mother prejudiced him against powerful women, fueling a lifelong penchant for imaginative, impressionable girls who would be unlikely to challenge him (McGlathery 1997, pp. 4–5). To be sure, when his parents' "unhappy" marriage failed, he was cruelly neglected by an emotionally distant mother who "withdrew to herself, so that he was raised more by his sisters" resulting in a "lack of parental guidance, authority, and affection [which] left Hoffmann, in childhood, rather more to his own devices . . . in the realm of his imagination."

Such background certainly informs Aurelia's plaintive moan: "Can there be anything more terrible . . . than to have to hate, detest, and abhor one's own mother?" (Hoffmann 1892, p. 460). None of this, however, excuses the paranoid misogyny in *Vampirism*, a tale in which "[m]ad passion or hysteria on the part of women serves as the object of terror experienced by young men in connection with becoming attracted to [them]" (McGlathery 1997, p. 107). An allegorical interpretation of its climax might arguably view it as an expression of the stereotypical masculine anxiety of watching one's pretty young wife age out of sexual fecundity, "become her mother", and—with the introduction of children—gradually eat away at her time, resources, peace of mind, and creativity. If accepted, such a reading—where Hyppolitus' insecure ego is consumed by "his guilt and revulsion about the connection between marriage and 'lust of the flesh'" (McGlathery 1985, p. 134)—displays a selfish and shallow devaluation of female humanity and a paranoiac distrust of the influence of older women as possible conspirators in a cabal of sexual corruption rather than members of a supportive nexus of feminine wisdom.

## 5. Sleepwalking Aurelia: Degenerate Dissident or Maturing Mother?

Aurelia's appeal to Hyppolitus stems from her obvious *variance* from her scheming, social-climbing mother. Where the older woman is mercenary, vulgar, and masculine, Aurelia codes as unmaterialistic, unspoiled, and unthreatening. Initially "attracted to her

by her melancholy beauty", Hyppolitus fosters a trust in her sincerity and docility because "she seemed far less eager for her marriage . . . than her mother", a hesitancy which is only compounded by her mother's death, which causes her to be even more "dependent on [Hyppolitus]" as a dominant protector and caretaker (McGlathery 1985, p. 134). But his discombobulating vision in the cemetery—real or imagined—exposes what he surely suspects to be a time-honored, intergenerational conspiracy between women, where simple-minded, adoring beauties are corrupted into conniving maneaters (figuratively and literally) by their avaricious elders. In time, both Aurelia and her mother become transfigured: they "increasingly assimilate themselves to the realm of the dead on account of their necrophagic tendencies", and—drawn by their appetites—they are transformed from something desirable into something unnatural and inhuman: "living persons who feed on corpses" (Sütterlin and Lambrow 2018, p. 115). Naturally, the moment of transition—from innocence to corruption—transpires at the moment of carnal understanding, and more specifically in the mysterious, liminal zone of pregnancy, which is a time when young women have historically relied more upon their female mentors (mothers, sisters, friends, midwives, doulas, and nurses) than their husbands for knowledge and emotional support. McGlathery asserts that by drawing these thematic parallels, Hoffmann conjures a cynical "connection between erotic passion and insanity, as manifested in a young woman's hysteria" (McGlathery 1997, p. 106), going so far as to unambiguously suggest that Hyppolitus merely "imagines that he has married the daughter of a ghoul", who is "only a fantasy projection of [his] subconscious guilt and panic about the prospect of marriage" (McGlathery 1985, p. 134).

Ultimately then, Hoffmann conceives of marriage as a bait-and-switch tactic where men are tempted into surrendering their sovereignty by mercenary gold diggers who are coached in tactics of sexual manipulation by their mothers. By sharing the techniques that were used against their own fathers, the mothers pass on the exclusive trade secrets of wifedom to their as-yet uninitiated daughters, and the cycle of spousal deception continues down the generations of women (or so Hoffmann would have it). Significantly, McGlathery discerns that "Aurelia's ghoulish compulsion . . . appears to have its origins farther back in her experiences resulting from her mother's affair", recalling its fatal impact on her *presumed* father ("the man she had always called 'Papa'"), her mother's subsequent sex work, and the Stranger's brutal regime of domestic violence and con-artistry (McGlathery 1997, p. 106). A trauma-informed eye would consider these episodes with profound sympathy, but Hyppolitus appears to view them as lessons in lifelong training in deception and disloyalty or as rites of passage intended to educate her on how to defraud, deceive, and generally con her way through hearts and into homes.

In this respect, whether his nightmarish vision is to be trusted or not, Hyppolitus fears that Aurelia can be likened to a ghoul—one who secretly gains entry into a sacred space (holy ground/matrimony) to feed on the vulnerable body of a man who naïvely entrusted himself to be allowed to rest there (in death/in marriage) without being violated. It is Hyppolitus' explicit fear—hounded as he is by "the darkest suspicions and forebodings" (Hoffmann 1892, p. 468)—that, like the desecrated dead man, he has foolishly entrusted his body, heart, and mind to rest peacefully in his marriage bed, oblivious to the humiliating infidelities and atrocities his wife is committing while he is vulnerably supine. Once confident that he has married a suitably appreciative and obedient ingénue, Hyppolitus finds himself shamed, enraged, and overwhelmed "with the wildest fury" (Hoffmann 1892, p. 470) to discover that, in the process of becoming pregnant, his young wife has metamorphosed into her mother, adopting "her . . . depraved way of looking at things" (Hoffmann 1892, p. 465), and allowing her conniving spirit to live on in their marriage; in effect, achieving her original failed machinations to infiltrate Hyppolitus' noble family by way of his wiser father:

> The mother desires Aurelia's marriage to [Hyppolitus] as compensation for her own disappointed hope of marrying [his] father. Likewise, Aurelia's nocturnal visits to the graveyard later, when she is pregnant, may have to do with her

fear that the mother, who according to Aurelia had tried to pander her to the executioner's son, would return from the grave to oppress her anew. At any rate, it is the mother's ill treatment of her, whether real or imagined, that is the object of Aurelia's hysteria in her marriage... (McGlathery 1997, p. 106)

## 6. Strategies of Social Support: The Subversive Culture of Pregnancy Cravings

One of the story's most bizarre and unique details is that Aurelia's vampirism is not brought on by the marriage itself, nor even by its consummation, but as an expression of pregnancy cravings. Selective hunger during the first trimester is often spoken of anecdotally in comical terms, especially as regards the unusual combinations of food that pregnant people insist on having on hand. In one medical study of the phenomenon's causes, which remain unknown, Caitlyn Placek remarks that: "[a]cross cultures, a craving for items not typically desired is often considered a hallmark of pregnancy. Women are known for craving sweets, fruits, calorie-dense foods, odd combinations such as 'pickles and ice cream', or pica substances, such as clay and chalk" (Placek 2017, p. 1). Just as in Hoffmann's day, the sudden urge to eat extraordinary foods or large volumes of food during pregnancy is mysterious, and it "remains relatively understudied", though the leading evolutionary theories "include a need to seek foods to either satisfy the energetic demands of the growing fetus, or to replenish nutrients lost from nausea, vomiting and food aversions in the first trimester" (Placek 2017, pp. 1–2). Intriguingly, these cravings are understood to be relative to the culture of the woman in question, with the types of foods craved, appetite triggers experienced, and levels of social support felt being uniquely "frame[d]" by their respective "cultural and environmental niches" (Placek 2017, p. 11). Furthermore, particularly toxic or taboo cravings have been hypothesized to "function as a social bargaining strategy among women who have higher fertility, feel pressure to have a son, heightened resource scarcity and experience psychological distress" (Placek 2017, p. 3).

Though selective hunger may be a mere medical curiosity to obstetrics researchers, it serves as a profound bonding trope of maternal culture—a relatable memory that can allow mothers to simultaneously experience the uniqueness and collectivity of their shared experience, i.e., a form of "emotional eating" that may "function as a signalling strategy for social support" (Placek 2017, p. 2). For Hoffmann, however, this natural shift in appetite is a symptom of motherhood's visceral abnormality and the perceived, alien grotesqueness of the female reproduction system itself. Hoffmann's exceptionally insensitive representation of Aurelia's pregnancy experience begins with the doctor's visit, during which he recklessly relates the story of the blacksmith's wife who murdered her husband in a fit of hormonal rage, an anecdote that serves to delegitimize, infantilize, and dismiss Aurelia's very *real* emotional state. As McGlathery observes, her cannibalistic "compulsion is seemingly *triggered* by her physician's talk about unnatural appetites and compulsions [emphases added] of pregnant women" (McGlathery 1997, p. 106), although whether Aurelia or Hyppolitus himself is more "triggered" by this conversation is subject to debate. Without responsible medical guidance—having been primed to suspect Aurelia of unstable irrationality and view her pregnancy with disgust and suspicion—Hyppolitus concludes that pregnant women are not to be believed, respected, or trusted because they will eventually be betrayed by their impulses and either become "wildly desirous or resent having been impregnated or both."

Like so many of Hoffmann's horror motifs, the idea of cannibalistic pregnancy cravings is rich with psychoanalytical suggestion. It examines a variety of common marital anxieties, chief among which is the worry that motherhood's acquired responsibilities will inevitably cause the male sexual partner to be demoted in the mother's list of priorities, even before birth. Seen through this interpretation, Hyppolitus is experiencing the archetypal sexual anxieties of new fatherhood, wherein he senses the end of the honeymoon phase and the cessation of his unchallenged primacy in his wife's life. Her appetites are no longer driven by her husband's needs but by those of the fetus inside her (and the future she desires for both her and it), and her husband finds himself burning with Freudian-level jealousy

for the invisible presence that is currently superseding his monopoly on her sexual being. Worse yet, by growing inside her enigmatic reproductive system, this unseen stranger has automatically exceeded its father in proximity to and intimacy with its mother's core sexual mystique. What could be seen as a mystical and almost divine relationship between mother and child is perverted into a jealousy on the part of the father where the unborn child is cast in the role of secret lover and the father in that of cuckold. With such a cynical point of view, it is no wonder that pregnancy cravings generated by conception are made analogous to the sexual longing of an adulteress. Indeed, Hippolytus consistently connects her lack of appetite (and later, her consumption of human carrion) with the erotic yearning of an assumed affair, directly correlating her "strange condition" with raging suspicious of her "conjugal infidelity" (Hoffmann 1892, p. 468).

Pregnancy cravings are not the only physical manifestation of the female reproductive experience that Hoffmann codifies as "Other" and obscene. Hyppolitus' first encounter with the Baroness is memorable for the moment when she crushes his hand while overcome by a "tetanic spasm" (an involuntary convulsion)—a regular tic by which she is regularly seized (Hoffmann 1892, p. 457). After her mother's death, Aurelia recalls the latter's parting curse which linked these seizures to the contractions she suffered during childbirth: "You are my misfortune, horrible creature that you are! But in the midst of your imagined happiness vengeance will overtake you, if I should be carried away by a sudden death. In those tetanic spasms, which your birth cost me, the subtle craft of the devil—" (Hoffmann 1892, p. 466). Although Aurelia cannot finish the sentence, begging Hyppolitus to "spare her the complete recital" of her mother's oath, Hoffmann rhetorically fuses the mysterious power of uterine contractions with the "subtle craft of the devil"—*viz*., the "subtle craft" of female sexuality. What else could explain how a well-bred woman—one like "noble, simple, and child-like" Aurelia (Hoffmann 1892, p. 458)—could be overtaken by such an explosive force emanating from within her own being? How could a passive dependent like Aurelia be responsible for the stunning might of birth spasms without any external influence? Hoffmann, Hyppolitus, and the baroness all seem to agree that there is something inscrutable and unsettling about this hidden might lurking within such an otherwise submissive young woman—a hidden source of internal agency that defies the comforting narrative of female fragility, casting yet further suspicion on the liminal zones of female sexuality and reproduction, where vulnerability and strength intermingle in shadowy, indefinable ways. There is so much more to Aurelia, he comes to realize, but these secret depths only gin up his insecurities, transforming her into a menacing new Creature.

Ultimately, Hoffmann represents his pregnant antagonist completing a transformation common to observational comedy, yet with the funereal self-seriousness of a Greek tragedy: she turns into her mother—metamorphosing from a desirable, virginal Madonna into a degenerate, vampiric Whore, (or, more accurately, from defenseless dependent to worldly woman). It is notable that Aurelia only assaults a living person, her husband, "after he exposes her abnormal craving" (Kamla 1985, p. 236)—*viz*., her taste for human (specifically men's) flesh. Beyond the surface-level taboo of cannibalism, this "abnormal craving" may also symbolize a woman's repressed hunger for parity with her husband in terms of independent worth and personal sovereignty. Disturbingly, as Barber recounts, Hippolytus' association between pregnant women and vampirism was not remotely unique to Hoffmann's story. Historically speaking, "pregnant women are viewed as especially apt to become revenants" (Barber 1988, p. 138), for "[a]lthough vampires are far more often male than female, the exceptions to the rule are commonly mothers who have died in childbirth" (p. 36). This prejudice, then, would seem to have an even broader, cultural basis founded in a distrust in the mystery of motherhood, proprietary to women and distrusted by men. Recalling Lerner's argument that women have historically earned subversive power through their "essential" monopoly on procreation and motherhood (Lerner 1979, pp. 3–4), it is telling that Aurelia's transformation from unthreatening trophy wife to savage cannibal coincides with the advent of her pregnancy—a development that opens her up to the uniquely female mysteries of human gestation and generation, indelibly separating her

from the dominion of her husband's authority by ushering her into a secret society into which he has no hope of ever being accepted. Note how he is particularly goaded by the notion that she is no longer content with the food of his table and has instead obviously taken it upon herself to secretly furnish her own sustenance, experiencing "the utmost repugnance" at the provisions he offers her (Hoffmann 1892, p. 467).

In this reading, Aurelia's lack of appetite is analogous to sexual rejection, informed by the patent "psychology of sexual revulsion . . . evident in her" attitude towards Hyppolitus following her pregnancy (McGlathery 1985, p. 134). Food was once a source of power he held over her, since when they met she was "almost starving" (Hoffmann 1892, p. 457), but after marriage and pregnancy—now endowed with legal protection, awakened by sexual self-awareness, and inducted into the secret mysteries of motherhood—she seems to have liberated herself from her husband's protection by uniting herself with the sisterhood of the cemetery, who coach her in subversive autonomy by way of ritualistic necrophagia. From Hyppolitus' paranoid perspective, he surely finds newfound significance in the story of her earliest childhood memory: looking down upon "this man she had always called 'Papa'" "laying stretched on a long table" and kissing his "cold as ice lips" (Hoffmann 1892, p. 461). He may infer that she views the roles of "husband" and "father" as mere temporary titles conferred provisionally rather than eternally, and worry that one day his own unborn child may casually refer to him merely as "this man I always called 'Papa.'" He may also interpret Aurelia's macabre, parting kiss to her father as both a domination of his corpse (as it lies stretched out in this supremely vulnerable posture) and a flirtatious hint at the many ugly, unnatural appetites of which he suspects her: "unconscious oedipal drives . . . desiring incestuous union . . . sexual domination and sadistic aggression" (Kamla 1985, p. 240), adultery, cannibalism, and vampirism.

Meanwhile, from Aurelia's perspective—assuming in this instance that Hyppolitus' graveyard vision was the product of his paranoid imagination—her very real and documented assault on him may have stemmed less from rage at the exposure of her true nature, and more from heartbroken despair at the thought of Hyppolitus' distrust in her. Such an interpretation would consider her gruesome, final moments as the manifestation of a self-fulfilling prophecy. McGlathery considers this more charitable reading, suggesting that: "when [Hyppolitus], following his nightmarish, patently fantastic graveyard vision of Aurelia in a crowd of female ghouls, accuses her and she reacts by sinking her teeth into his chest, she may be seen as acting out of hysterical despair that he could think her capable of such a reprehensible practice" (McGlathery 1997, p. 107). In either case, her story ends in tragedy. After a life spent surviving the traumatic abuses of her mother and her rapacious pimp, she falls prey to her own husband's neurotic distrust before she is allowed a chance to reverse the generational trauma of her upbringing. She expires with her unborn child in the futile act of rebelling against society's expectations that she allow herself to be used, defined, and limited by the powerful people who see fit to use her body for their own ends. While pregnancy may be limiting to some people, for Aurelia, it was arguably her first experience of autonomy. As a means to escape the manipulations of her mother, the Stranger, and Hippolytus, pregnancy allowed her to experience an original rite of passage unique to herself in which she hoped to finally become initiated into the self-determination and subversive wisdom of motherhood, an exclusive domain safeguarded by women and envied by men. Hippolytus, on the other hand, who shares the author's sexual frustrations, proves himself to be a paranoid misogynist, consumed with conspiratorial anxieties about the one sphere his gendered, financial, and social power can never encroach upon. While he suspected his wife of adultery, the reality seems to have been his worst nightmare. Rather than seeking succor from another man, he finds her emancipating herself at the feet of a group of female mentors. For adultery she would surely have been shamed and divorced, but for this more serious crime—that of defying the patriarchal order in which Hippolytus defines himself and all reality—she must sadly pay with her life and that of her unborn child.

### 7. Conclusions: Secrets, Conspiracies, and the Destructive Folklore of Social Distrust

An ever-expanding whirlpool of suspicion and secrecy forms *Vampirism*'s philosophical core, most obviously expressed in the deepening cycle of distrust that exists between its alienated husband and wife. But there is a further level of secrecy that provides the story with its reputation as an unsettling reading experience, i.e., the unresolved secrets to which the reader is ultimately denied access. The shifty nature of *Vampirism*'s many riddles insist on a complex interpretation and naturally require the reader to make a series of judgements based on the reliability of the narrator, the psychology of the characters, and the nature of its possible case of supernaturalism. To this final point, there are a number of reasonable interpretations to which readers could subscribe. In no particular order, some of the most significant theories are: that Aurelia is a literal self-aware vampire; that Aurelia was seen cannibalizing a corpse but as a result of mental illness or a curse; that Hippolytus experienced an allegorical vision or dream exposing her duplicitous nature; that Hippolytus suffered a hallucination, onset by his building paranoia and insecurities; that Hippolytus never saw any of this at all but fabricated the story as a cynical test of character; that Hippolytus fabricated the story out of pure hatred, simply to punish his traumatized wife; and finally, that the entire story—or elements of it—is an exaggeration, a parable, or a rhetorical invention meant to flesh out the otherwise pedestrian story of a disastrous marriage.

The seismic struggle between truth and deception forms the fault line on which Aurelia's marriage teeters. Several basic questions about her nature, activities, and destruction must be probed in order to better understand what this story can tell us about the motives and perils posed by speculative narratives of distrust. The most basic question we must ask is whether or not we believe that Aurelia is a cannibal at all. The implication of this inquiry webs out into a wide range of varying conclusions that change the reader's understanding of her character and the story's essential message. Hippolytus appears to believe what he claims to have seen, and his anxieties stem from a conspiracy theory that Aurelia is a kind of sleeper cell planted by her mercenary mother—a honey trap sent to infiltrate and gradually overtake his family power. This interfaces with the female vampire's historical role as a scheming outsider intent on subverting the dominant social order—an insidious usurper who must be destroyed largely because her "open sexuality becomes a threat to the . . . community" (Holte 1999, p. 170).

If we choose to proceed with the assertion that Aurelia is a genuine ghoul, the next question must be whether her impulse to cannibalize is conscious or unconscious. Is she able to resist the urge, or is it something that completely overpowers her will, perhaps even without her awareness? This raises the inevitable specter of mental illness—yet another form of suspected "Otherness" that historically permeates Gothic tropes from *The Tell-Tale Heart* to *Jane Eyre* (Khair 2009, p. 16). Should those who suffer from inherited mental illness be considered guilty or sick? Culpable or treatable? Should they be cast into prisons, strapped in asylums, burned at the stake—or should they be approached with the compassion we would extend to a person battling a genetic disease? Even if Aurelia is a literal vampire, we must be highly skeptical of Hippolytus' approach. Did she fully realize what she was doing, or has she been unconsciously driven by her charnel pregnancy cravings? Has she been sneakily raiding the cemetery for a midnight snack or is this a gruesome example of sleep-eating? The implications of each possibility to this mystery once again take us off into different directions. Even if Aurelia was conscious of her cannibalism, how much of her appetite was analogous to addiction rather than desecration? Or, even if she had full knowledge of her activities, and even if she could have done more to resist her compulsions, how much of her cannibalism was the cause of an insidious conspiracy versus an inherited compulsion?

If we suspect the former, then perhaps this story rightfully warns men about the importance of vetting their mates, but if we surmise the latter—regardless of Hoffmann's own sexist prejudices—then we must question why Aurelia was not allowed a chance to find treatment, support, or understanding. Why was she publicly exposed and humil-

iated like a captured burglar rather than privately taken into her husband's concerned confidence? The answer, I would argue, lies with her gender, and the lack of forgiveness that early nineteenth century Europe afforded to women whose emotions, sexuality, or neurodivergence bled out beyond the privacy of their interior life, founded on the dehumanizing belief that women "participated in the hierarchy only as daughters and wives, not as individuals" and were expected to compliantly "occupy an inferior and subordinate position" (Lerner 1979, p. 12). If a woman was unfortunate enough to have her personal trials impact her ability to perform her highly regulated social role (e.g., wife, mother, sister, daughter, governess, laundress, maid, prostitute, patroness, or churchwoman), then she would not have been afforded the grace that—ideally—we would extend to a struggling partner, friend, or colleague today. Hoffmann's contribution to the folklore of misogyny, if read through this feminist lens, arguably reinforces traditional suspicions of female agency up to and including the fundamental maternal impulse to care and provide for her unborn child over the needs of her fully grown husband.

On the other hand, if Hippolytus' experiences are imagined, what unresolved secrets could they point to—and whose secrets? If the secrets belong to Aurelia, could they be clues—unreliable though they may seem—to some hidden reality of her character? Could the Baroness' conniving spirit be dormant in Aurelia, waiting to be awakened by motherhood, whether or not she is a literal ghoul? If so, is Hippolytus' conspiracy theory, like those that unnecessarily complicate genuine tragedies (e.g., JFK's Assassination, the Boston Marathon Bombing, or the 9/11 Attacks), propagating comforting fantasies and approachable "political explanations" to conveniently vilify "the forces that shape ... political culture" (Oliver and Wood 2014, p. 952)? If so, then Hippolytus' complex narrative of Aurelia's vampirism simultaneously does a disservice to his innocent wife and distracts from any genuine character issues she may have and could have been addressed with patience and compassion. However, if the secrets belong to Hippolytus, then the entire nature of his vision must be called into question. If, for instance, Hippolytus' vision is a complete fabrication, we must completely reimagine the narrative. It is no longer about a duplicitous femme fatale or a man's marital insecurities but about a woman who is repeatedly and traumatically used by her toxic loved ones—a woman who cannot escape the cycle before she becomes a victim of her husband's misogynistic paranoia.

If—as I assert—this is true, we must then ask ourselves how much of Aurelia's descent into madness and death stems from any inherited mental illness, and how much of it is directly caused by Hippolytus' gaslighting? The second possibility deserves very serious consideration, especially from a reading that is concerned with the deadly effects of conspiracy theories and social distrust along gender lines. Although each of the many interpretations of this story bear consideration, Hippolytus' pattern of impulsivity—his sudden attraction to and engagement with Aurelia, and his just as sudden transition from trust to suspicion—is sufficient to call into question his reliability. As his paranoia mounts, Hippolytus' self-control is increasingly overwhelmed by "the darkest suspicions and forebodings", and Hoffmann directly invites the reader to consider "into what a state of mind" the Count's escalating anxieties put him (Hoffmann 1892, p. 468). With his objectivity deductively and explicitly called into question, the nature of Hippolytus' outlandish vision can be responsibly called into question. We are free to consider the possibility that he has invented it—consciously or unconsciously—as a wish fulfillment, providing a self-justifying excuse for him to turn against his wife as her attention gradually shifts from him to the child she is gestating.

If such is the case, we are likely to be reminded of far more sinister conspiracy theories, such as those with no germ of truth, the sort that are fabricated as plausible excuses to allow atrocities to be committed against marginalized communities. Indeed, this connects us back to the very nature of "female vampire" folklore and the literary trope it inspired among the Romantics. Like Hippolytus, post-Revolutionary Europe was reeling with political insecurity and a desire to be restored to normalcy after the "swath of devastation cut across Frane" and "the disillusionment and terror that followed" (Paulson 1981, p.

545). Instead of investigating the causes of this social upheaval with much needed soul-searching, the nineteenth century bourgeoisie found comfort in an evasive conspiracy narrative that they were victims of rapidly progressing minorities (women in particular) and that the source of this malaise was the threat of women's unregulated wills and bodies, a suspicion that stemmed from a mainstream "concern—even obsession—with women's actual power, an obsession that increased as the century progressed" (Senf 2013, p. 154). In *The Vampire in Nineteenth Century English Literature*, Carol A. Senf quite capably makes the argument that these sexist anxieties about female sexuality and reproduction were directly responsible for "the increasing popularity of fatal women (of which the vampire is merely one important sub-type)" as "women gained power and influence" throughout the century, causing "concerned ... writers in the nineteenth century [to respond] by creating powerful women characters, the vampire being one of the most powerful negative images" (Senf 2013, p. 154).

A woman like Aurelia, then, who is suspected of having desires outside her husband's bed, is automatically cast as a villain without consideration for her abusive childhood and history of sexual abuse. Indeed, many critics, Jameson among them, have noted that the Gothic trope of the woman fleeing from ravishment disguises its savage sexism with gaslighting pearl clutching:

> Certainly the gothic mobilizes anxiety about rape, but its structure gives us the clue to a more central feature of its content ... its classical form turns on the privileged content of the situation of middle-class women—the isolation, but also the domestic idleness imposed on them by newer forms of middle-class marriage. (Jameson 1991, p. 289)

In short, a wife was just another personal possession, like any other decorative object d'art in her husband's parlor, and her personal and sexual fulfillment was seen as a subversive threat to his (and his society's) equilibrium—certainly not as a pursuit worth encouraging. Hence the need arose for a literary folklore to develop around the secret cabal of women's private spaces and societies—midwives, girlfriends, female support systems—which conveniently interfaced with the extent European folklore of vampirism. Like vampires, women (...women wanting to marry, women who were pregnant, women with pregnancy cravings, women with children, women with needs of any stripe) posed a threat to their suitors and husbands. They were yet another insidious, subversive conspiracy intending to topple their male autonomy and challenge the heterodox values of their jealously patriarchal society.

The final secret—and the one that is most literally a secret (in the sense that we know for a fact that it consists of purposefully withheld information)—concerns the nature of the Baroness' enigmatic parting words: "You are my misfortune, horrible creature that you are! But in the midst of your imagined happiness vengeance will overtake you, if I should be carried away by a sudden death. In those tetanic spasms, which your birth cost me, the subtle craft of the devil—" (Hoffmann 1892, p. 466). Since the hex is left incomplete and we are provided with no obvious clues as to their possible composition, we are forced to conjecture based on Aurelia's own history and character. What, we must wonder, would a young woman with her traumatic past be so terrified to face that she could not even muster the strength to repeat the words? Several possibilities jump to mind, with the most literal interpretation being that—should the Baroness not be allowed to enjoy the perks of the marriage she has brokered—Aurelia will magically inherit her mother's vampirism. Another more psychoanalytical interpretation might focus on the final words' reference to the correlation between birth contractions and treacherous devilry, i.e., childbirth and motherhood might be said to naturally bring with them a secret curse. What, exactly, this mystery might be is once again the subject of our speculation. As previously argued, it may be that the Baroness has informed Aurelia that the ritual indoctrination of pregnancy and parenthood cause a daughter to "become" her mother. This would be just cause for Aurelia to choke on the mere words, for—as she says—she "hate[s], detest[s], and abhor[s her] own mother" (Hoffmann 1892, p. 460).

The prospect of adopting her mother's detestable character would naturally terrify the woman who had just so recently escaped the gravitational influence of her parent's abuse. Even here though, we are left to wonder what elements of the Baroness' personality Aurelia might most fear adopting: her reckless sexual appetite (awoken by the loss of her virginity); her lack of self-respect (raised by her same deflowering); her helpless dependence on men (caused by her arranged marriage); her lack of financial and social freedom (exacerbated by her impending motherhood); or her relentless obsession with finding her unborn child male protection (should it be hapless enough to be born female)? Her mother's traumatic legacy overshadows Aurelia from her first memories, through her emancipation, and up to the moment of her ignominious death. For Hippolytus, the enigmatic "subtle craft of the devil" is implied to be the loss of virginity, i.e., Aurelia's perceived transformation from Virgin to Whore after the alchemy of sexual congress. For Aurelia, however, the subtle craft in question is the transformation from girl to woman and from woman to mother, and the curse of this subtle craft is in some sense the legacy of generational trauma—the cycles, vices, and sad fates of earlier generations tend to be repeated on into the future, passed from mother to daughter. She has been cursed to suffer as her mother has suffered at the hands of violent men who will commodify her body and mortgage her soul as their own property.

Indeed, it is worth noting that this culture of mystery and distrust is not original to the unhappy couple, but an example of the subtle craft of generational trauma. Both parties grew up with paranoid guardians whose relationships (*viz.*, their children and members of the opposite sex) were spoiled with secrecy. Psychological studies have repeatedly demonstrated that "people with unusual levels of anxiety, paranoia, or personal mistrust are also likely to be attracted to conspiracy narratives", causing them to "see enemies everywhere" (Oliver and Wood 2014, p. 954). Aurelia and Hippolytus certainly embody these antisocial traits, which likely stem from their adverse childhood experiences. These inherited traumas trained Aurelia and Hippolytus on what to expect from close relationships and how best to navigate them to avoid pain. Sadly, this proved to be a plan for disaster. For Aurelia, a web of probable conspiracies entangled her entire childhood, beginning with the cause and circumstances of her father's mysterious death; the identity of the Baroness (whom she never met until his passing); the intentions of the Stranger for mother and daughter alike, and the uncertain level of danger he posed to each; the nature of the relationship between the Stranger and Baroness; and the Baroness' involvement in Aurelia's arranged marriage, including her motives, expectations, and, above, all her mysterious past with Hippolytus' father. The latter character's lack of transparency (specifically, his unwillingness to be direct and vulnerable with his son regarding his relationship with the Baroness, the reasons for his distaste, and the potential dangers posed by an alliance with her family) encourages the younger man to allow his imagination to run wild with elaborate suspicions. Without clear communication, Hippolytus—like Aurelia—is inclined to repeat his parent's mistakes. Unwilling to trust his spouse, he finds comfort in his suspicious fantasies and allows the folklore of social distrust to exacerbate his inherited paranoia.

Both partners come away from their respective families with an intrinsic suspicion of others (especially members of the opposite sex) and Hippolytus, in particular, steers blind into his marriage, finding disordered relief in his unrestrained conspiracy theories, whereby he finally achieves the level of understanding and competence denied to him by his secretive father. Both spouses are also suspicious that the *ultimate* conspiracy in their massive catalog of suspicions is their marriage itself. Hippolytus obviously suspects Aurelia of colluding with her mother to subvert his family line, but Aurelia just as surely must wonder to what extent Hippolytus has conspired with her mother to bring her under the tyranny of his increasingly abusive power, repeating the cycle she had so hoped to escape. Her violent reaction to his accusation may have been, as previously argued, a hopeless response to her horror at having fled the secretive cell of her mother's machinations only to find herself legally locked into another insidious conspiracy of repression.

Ultimately, Hippolytus and Aurelia devote themselves to nourishing a culture of distrust and secrecy that dooms their marriage to failure. Their ever-escalating chain reaction of jealous assumptions fully metastasize into convincing conspiracies that provide each of them with a comforting narrative designed to protect their egos. However, while Hippolytus' suspicions are founded in misogynistic insecurities handed down by his secretive father and reinforced by a repressive society, Aurelia's distrust of her husband is grounded in the repeatedly reinforced evidence of her personal history. Her final assault on her husband comes with the sudden, definitive reactivity of a person who has been on guard against outside attackers for longer than she can remember, as if she has always been prepared for her husband to transform into an enemy and is ready to lash out at him in self-defense at a moment's notice. Unfortunately, *her* conspiracy theory proves a reality. The paranoia she has carried with her—the fear that she will once again fall into the hands of toxic abusers—is realized when her husband exposes her secret (whether real or imagined) and turns her over to public shame instead of supplying her with privacy and understanding. Like all conspiracy theories, which often develop when people's "existential needs are threatened, as a way to compensate for those threatened needs", hers centers around a fear of having her agency taken out of her hands by a secret collusion between nefarious forces (Douglas et al. 2019, p. 8). It is her misfortune that this traitor is none other than her husband, and although we are not given the entirety of her mother's parting curse, there is no doubt that her prophesized calamity has come to pass.

Hopelessly tangled up in their respective patterns of cynicism, both partners are destroyed through the inescapable gravity of their own obsessive paranoias and the intoxicating comfort of allowing their suspicions to compound in their imaginations. Incapable of reaching out in vulnerability and trust, the couple are lost to each other and, ultimately, to their own selves as they descend into insanity—a self-soothing state in which their worst fears are realized and validated, regardless of their veracity. Their deaths speak directly to their reliance on secrecy for protection and the resulting infestation of conspiratorial thinking that sucks them down into black holes of exponentially compounding anxiety. And yet, their causes for secrecy are obviously completely different. Aurelia's guardedness is proven justified as, indeed, her husband has proven untrustworthy, while the rationality of Hyppolitus' paranoid plotting remain a matter of conjecture. Was he vindicated in his suspicions, or were they the result of an insecure and prejudiced mind? While Hoffmann himself may have shared Hippolytus' suspicion of the secret culture of pregnant women, the actual story he wrote leaves us without any clear resolution and allows us to wonder at the degenerative power of conspiratorial ideation and the dangerous (and in this case *deadly*) folklores of social distrust that all societies harness to the "Other", their most vulnerable members, while excusing the unconscionable abuses of the cultural majority.

**Funding:** This research received no external funding.

**Conflicts of Interest:** The author declares no conflict of interest.

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
