# Peer review of "“The Subtle Craft of the Devil”: Misogynistic Conspiracy Theories and the Secret Society of Pregnancy Cravings in E. T. A. Hoffmann’s Vampirism"

_humanities, doi:10.3390/h12060143_

Round 1
Reviewer 1 Report
Comments and Suggestions for Authors
This article offers a feminist reading of Hoffman's 'Vampirism'. The first four pages provide a summary of early Gothic and vampire fiction: this material is now well-known and can be condensed. Pages 5-6 offer a summary of the tale itself: again, this can be condensed, as the original story is readily available.
The main content of the article is the interpretation offered in pages 7-13. The feminist interpretation of the story as evoking male anxieties over female sexuality, pregnancy, and childbirth is entirely plausible, but the argument is weakened by a persistent tendency to describe and judge the characters as though they were real people, and to simplify material which gains its whole strength from its inherent ambiguity. If the story was simply and straightforwardly a misogynistic tale about how innocent girls become manipulative, life-draining 'vamps' after marriage, then it would hardly be worth studying in any detail: its power comes from the fact that so many of the key elements in its story are allowed to remain mysterious. (Is Aurelia's cannibalism conscious? Is it voluntary? Did the scene in the graveyard really happen? What did her mother say to her, in the speech that she cannot bear to repeat?)
The whole horror of the story comes from the fact that no explicit explanation is provided for Aurelia's apparently unmotivated pivot from virtuous virgin to cannibal ghoul, a shift that no realist psychology could possibly fill: it runs on the logic of nightmare, and it is as nightmare that it must be understood. None of this is to say that the basic thrust of the argument - that the story gains its charge from Hoffman's simultaneous attraction and repulsion to women in general, and female sexuality in particular - is incorrect: but a richer analysis will result from respecting the story's silences and ambiguities, rather than attempting to flatten them out!
I would also recommend that the essay makes a clearer statement regarding its own originality. Feminist readings of vampire fiction are now very numerous, and it is thus important for any new addition to the genre to make clear what new insights it adds to the existing scholarship.
Comments on the Quality of English LanguageEnglish is mostly fine but with some issues, e.g. '18th' and '19th' instead of 'eighteenth' and 'nineteenth', and 'it's' instead of 'its'.
Author Response
Many improvements have been made based on your very helpful and insightful advice. Please see the attachment.
My very heartfelt thanks to you for your help with this!
Michael Kellermeyer

Reviewer 2 Report
Comments and Suggestions for Authors
The chief contribution of the essay comes through its examination of a little known vampire story by E. T. A. Hoffmann. The essay situates this text within vampire literature more generally and usefully notes some of the differences between the German and the English Gothic traditions. There is also a fruitful discussion of maternity and pregnancy cravings in the story, linked to vampire fiction’s anxieties regarding female sexuality. The author reads the story as misogynistic in its representation of Aurelia as a woman who in pregnancy comes to resemble her monstrous mother, linking this representation to Hoffmann’s own prejudices against women.
The essay is clearly written and structured, with very few typos etc. However, N.B. that the Count’s name is spelled inconsistently as Hippolitus and Hippolytus and a reference is missing on l. 548. The abstract commences with a sentence fragment.
The main revision that I would recommend relates to the conclusion. The essay ends rather abruptly without a proper conclusion and would in my view benefit from a conclusion returning to the story’s place in the overall vampire tradition, which rather falls away during the latter stages of the analysis.
Author Response

(The authors gave the same response as above.)
